# LcTprxII Overexpression Enhances Physiological and Biochemical Effects in Maize Under Alkaline (Na_2_CO_3_) Stress

**DOI:** 10.3390/plants14101467

**Published:** 2025-05-14

**Authors:** David Pitia Julius Michael, Qing Liu, Yuejia Yin, Xuancheng Wei, Jainyu Lu, Faiz Ur Rehman, Aroge Temitope, Buxuan Qian, Hanchao Xia, Jiarui Han, Xiangguo Liu, Long Jiang, Xin Qi, Ruidong Sun, Ziqi Chen, Jian Zhang

**Affiliations:** 1Faculty of Agronomy, Jilin Agricultural University, Changchun 130118, China; pitialotoro@gmail.com (D.P.J.M.); lu@mails.jlau.edu.cn (J.L.); faizurrehman538@gmail.com (F.U.R.); aroge.ttop@gmail.com (A.T.); qqbx117@163.com (B.Q.); lxgyyj@cjaas.com (X.L.); xinq@jlau.edu.cn (X.Q.); sunruidong@jlau.edu.cn (R.S.); 2Institute of Agricultural Biotechnology/Jilin Provincial Key Laboratory of Agricultural Biotechnology, Jilin Academy of Agricultural Sciences (Northeast Agricultural Research Center of China), Changchun 130033, China; liuqing892@163.com (Q.L.); weixuancheng2022@163.com (X.W.); xhczd0315@163.com (H.X.); hdrplybest@163.com (J.H.); 3Institute of Agricultural Quality Standard and Testing Technology, Jilin Academy of Agricultural Sciences (Northeast Agricultural Research Center of China), Changchun 130033, China; yyjqishi@163.com; 4College of Agronomy, Jilin Agricultural Science and Technology University, Jilin 132101, China; jlnykjxyjl@163.com

**Keywords:** LcTprxII, alkaline stress, type II peroxiredoxin, transcriptome, hormone regulation

## Abstract

Alkaline stress limits crop productivity by causing osmotic and oxidative damage. This study investigated the new gene LcTprxII, a type II peroxiredoxin encoded by Leymus chinensis, and its role in enhancing alkaline stress tolerance in transgenic maize. The gene was cloned, overexpressed, and characterized using RT-PCR, phylogenetic analysis, and motif identification. Transgenic maize lines were generated via Agrobacterium-mediated transformation and subjected to physiological, biochemical, and transcriptomic analyses under alkaline stress. Under alkaline stress, the results revealed that LcTprxII overexpression significantly preserved chlorophyll content, mitigated oxidative damage, and maintained growth compared to wild-type plants, as evidenced by elevated activities of antioxidant enzymes (APX, CAT, SOD, and POD) and reduced malondialdehyde (MDA) content. Transcriptomic profiling identified 3733 differentially expressed genes and the upregulation of ABA and MAPK signaling pathways, highlighting the role of these genes in stress signaling and metabolic adaptation. Hormonal analysis indicated reduced ABA and increased GA levels in the transgenic lines. This study identified WRKY, bHLH, and MYB transcription factors as key regulators activated under alkaline stress, contributing to transcriptional regulation in transgenic maize. Field trials confirmed the agronomic potential of *LcTprxII*-overexpressing maize, with yield maintained under alkaline conditions. The present study revealed that LcTprxII enhances antioxidant defenses and stress signaling, which trigger tolerance to abiotic stress. Future studies should explore the long-term effects on growth, yield, and molecular interactions under diverse environmental conditions.

## 1. Introduction

Infections and alkaline salts, as biotic and abiotic stresses, severely affect crop growth and agricultural output worldwide. The total area of saline–alkaline soils worldwide exceeds 434 million hectares [1,2]. Moreover, alkaline stress is recognized as a more detrimental factor than salt stress. It significantly impairs plant growth and development, making it a critical concern for crop productivity [3]. Over 70% of the land areas that have undergone varying degrees of alkalinization continue to expand in northeastern China [4]. Chinese alkaline soils contain substantial amounts of (Na_2_CO_3_) and (NaHCO_3_) [5]. Alkaline and salt affect plant development and growth via osmotic variables, ionic toxicity, dietary abnormalities, and ROS bursts [6].

The two-phase model of alkaline damage outlined indicates that plant development first decreases because of osmotic stress and subsequently because of the accumulation of sodium ions (Na^+^) [7]. Nevertheless, the two stages are similar. However, because it increases soil pH, sodium carbonate (Na_2_CO_3_) has a more noticeable adverse influence on plants than neutral salts, such as sodium chloride (NaCl). This suggests that alkaline stress is typically more destructive to plants than neutral salt stress.

Alkaline stress induces significant amounts of different ROS, such as hydrogen peroxide (H_2_O_2_), superoxide (O_2_^−^•), hydroperoxide (ROOH), hydroxyl peroxide (HO_2_^−^•), and hydroxyl radicals (OH•). The rise of ROS levels can lead to oxidative damage to plant proteins and lipids [8]. This damage significantly restricts plant growth and maturity, resulting in wilting and plant mortality [9].

Plants have evolved complex systems to preserve redox homeostasis when challenged by adverse environments [10]. In response to environmental stress, plants minimize ROS using enzymes like POD, SOD, CAT, AsA, GSH, enzymes of the AsA–GSH cycle, and other antioxidants [11]. Recent studies have demonstrated the antioxidant functions of mercaptan reductase in plants [12].

Chinese wildrye (*Leymus chinensis*), a monocot perennial grass, flourishes in alkaline–sodic soil. It has developed methods to adapt to unique living conditions through evolution. There are reports that the expressed sequence tag (EST) of type II Prx in Chinese wildrye, which was treated with Na_2_CO_3_, was described to be ascending [13].

Peroxiredoxins (Prxs) are classified into four types according to their sequence homology and catalytic function: (a) 1-Cys Prx, (b) Prx II, (c) 2-Cys Prx, and (d) PrxQ [14]. Chinese cabbage was initially characterized as a type II Prx [15]. Subsequent genetic and biochemical studies revealed several type II Prxs in dicots. Studies have indicated that *Arabidopsis thaliana* has H_2_O_2_-reducing activity in type II Prxs [16]. Additionally, poplar type II Prx has been shown to lower H_2_O_2_, tBOOH, and CM-OOH levels in vitro [17]. Peroxiredoxins (Prxs) are involved in various cellular functions, such as providing antioxidant protection [18], facilitating H_2_O_2_-dependent cell signaling [19], and performing molecular chaperone roles [20]. These signaling pathways encompass abscisic acid (ABA) and jasmonic acid (JA) and influence the alkaline stress response by modulating the transcriptional activity of Prx II genes, although transcription levels remain unchanged with plant age [21,22]. Previous research on type II peroxiredoxin (Prxs) physiology, plant hormones, and biochemistry in maize (*Zea mays* L.), *Arabidopsis thaliana*, and rice (*Oryza sativa*) has demonstrated this and transcription regulation of plants [23,24].

Chinese-wildrye-growing plants have evolved responses that adapt to various environmental threats. Finding functional gene(s) primarily responsible for stress response could help improve crop yields via biotechnological breeding. In a previous study, a peroxiredoxin-like sequence was identified in *Leymus chinensis* that was Na_2_CO_3_-treated under simulated alkaline conditions [13]. We cloned a type II peroxiredoxin gene, *LcTprxII*, which is induced by alkaline stress and shows elevated antioxidant activity. In this study, we overexpressed the *LcTprxII* gene in maize to investigate whether it can increase maize alkalinity resistance and its regulatory pathways. This study provides a novel approach with the potential to improve alkaline stress tolerance in maize dicots and monocots.

## 2. Results

### 2.1. Phylogenetic Relationship and Expression Pattern of LcTprxII

To investigate the evolution of LcTpxII, a phylogenetic tree was constructed based on the similarities of the deduced amino acid sequences of 28 available type II Prx genes from various plant species (Figure 1A). Based on Dietz’s classification, the *Prx* proteins formed four major types, and *LcTpxII* clustered most closely with type II *Prx* members, including those from *Zea mays*, *Oryza sativa*, and *Arabidopsis thaliana*. Sequence analysis indicated that *LcTpxII* shares approximately 90% identity with *Zea mays* type II *Prx*, 87% with *Oryza sativa*, and 75% with *Arabidopsis thaliana*. The conserved cysteine residue at position 51 was identified as potentially involved in intramolecular disulfide bond formation during the catalytic cycle. Further sequence characterization of *LcTpxII* confirmed a 489 bp open reading frame encoding a 162 amino acid polypeptide with an estimated molecular weight of 17,400 Da and an isoelectric point of 5.12. Motif analysis revealed a thioredoxin domain and four predicted motifs: an N-myristoylation site, a casein kinase II phosphorylation site, an amidation site, and a protein kinase C phosphorylation site (Appendix A). Overall, the phylogenetic placement and high sequence identity with monocot *Prx* proteins strongly indicate that *LcTpxII* is a member of the type II *Prx* subfamily (Appendix A). A binary plasmid was assembled for plant transformation (Figure 1B). The *LcTpxII* coding sequence was placed under the control of the maize polyubiquitin (*ZmUbi*) promoter. At the same time, the bar selection marker was governed by the promoter of the Cauliflower mosaic virus 35S (CaMV 35S). A Nos terminator was positioned downstream of *LcTpxII* to ensure appropriate transcription termination. Following *Agrobacterium*-mediated transformation, transgenic lines (#1, #2, and #4) were verified, and the expression of *LcTpxII* was quantified relative to wild-type (WT) plants through qRT-PCR (Figure 1C). The expression levels of LcTpxII in each line were assessed using qRT-PCR. It was observed that the levels varied among the three lines, with line #1 exhibiting the highest expression, line #4 showing the lowest, and line #2 displaying an intermediate level. Notably, no expression of LcTpxII was detected in the wild-type (WT) samples. These findings align with expectations and confirm the normal expression of LcTpxII in the transgenic plants, confirming that the introduced construct effectively drove the overexpression of *LcTpxII* in the transgenic plants. Measurements of plant height are presented in Figure 1D. The WT plants reached a mean height of approximately 160 cm. In contrast, each of the transgenic lines (#1, #2, and #4) displayed reduced height, ranging between 120 cm and 130 cm. These decreases were statistically significant, with *p* < 0.001 for all three lines compared to WT. Among the transgenic lines, the smallest average height was observed in line #1 (approximately 120 cm), although no statistically significant differences in height were detected among the three lines themselves.

### 2.2. Physical and Biochemical Interactions of LcTprxII Overexpression with Alkaline Stress

To assess the impact of LcTprxII overexpression on alkaline stress tolerance, three T_2_ transgenic maize lines (#1, #2, and #4) were analyzed under alkaline-stressed conditions. The growth rate showed slight variations after reaching the three-leaf stage in a greenhouse with normal conditions. Still, no other noticeable phenotypic differences were observed between transgenic lines (#1, #2, and #4) and control plants (Figure 2A). Following nine-day exposure to 75 mmol/L of alkaline treatment, control plants exhibited visible phenotypic changes. In contrast, transgenic lines demonstrated reduced damage (Figure 2B). Under alkaline stress conditions, chlorophyll content and plant growth rate were monitored over time. Prior to alkaline treatment, transgenic maize exhibited higher chlorophyll content and shorter plant height compared to the WT. Upon exposure to alkaline stress, WT plants displayed a slower growth rate than transgenic maize (Figure 2C). The reduction in chlorophyll content occurred at a significantly slower rate in transgenic plants compared to WT (Figure 2D). These findings suggest that transgenic maize experienced less disruption in growth and development under alkaline stress than WT plants.

### 2.3. Effects of Alkaline Stress on Antioxidant Enzymes and Oxidative Markers

Antioxidant enzymes and oxidation markers are widely recognized as key indicators of plant response to environmental stress. These factors play critical roles in mitigating osmotic and oxidative stress, thereby contributing to tolerance to alkaline stress. These specific indicators were examined in the current investigation, with results presented in Figure 3A. Transgenics exhibited significantly lower levels of H_2_O_2_ compared to the WT. The levels in the transgenic lines were 0.10–0.01-fold, 0.05–0.01-fold, and 0.10–0.01-fold those in control plants. These results indicate that the overexpression of *LcTprxII* contributes to a reduction in the accumulation of this ROS. The MDA content of transgenic maize after 9 days of alkaline stress treatment was reduced by 20%, 30%, and 40% when compared to the WT (Figure 3B), leading to lower levels of oxidative damage to cell membranes. Antioxidant reductases, including APX, SOD, CAT, and POD, have been reported to be associated with the elimination of ROS. In the present study, compared to the WT, in APX activity, transgenic lines exhibit significantly lower reduction by 1.29- to 2.49-fold (Figure 3C), and the activity levels of SOD, CAT, and POD in transgenic maize increased by 2.38–2.51-fold (Figure 3D), 3.30–2.32-fold (Figure 3E), and 2.78–2.37-fold (Figure 3F), respectively, when under alkaline stress conditions. However, the results indicate that *LcTprxII* overexpression enhances antioxidant enzyme activity, reduces the accumulation of ROS, and decreases oxidative damage to the cell membrane in maize.

### 2.4. Differential Transcriptome Analysis Between Transgenic Maize and Control Plants

Differentially expressed genes from comparisons between transgenic and wild-type plants were analyzed using GO functional enrichment methodology.

First, 3733 DEGs were annotated within the Biological Process (BP), the Cellular Component (CC), and Molecular Function (MF) categories (Appendix A). Pathways related to metabolic and cellular processes, such as cellular metabolic processes (GO:0044237), are prominently enriched, emphasizing active biochemical and physiological functions. In the Molecular Function (MF) category, terms like binding (GO:0005488) and catalytic activity (GO:0003824) underscore the importance of molecular interactions and enzymatic processes. The Cellular Component (CC) analysis identifies the cytoplasm (GO:0005737) and other intracellular compartments as key loci for these activities, reflecting the centrality of intracellular dynamics. Furthermore, a KEGG analysis was performed in the present research. It was shown that the enrichment of 1793 DEGs in 14 pathways was coupled with alkaline stress. The plant–pathogen interaction and plant hormone signal transduction pathways are the most enriched, indicating a strong emphasis on defense signaling and hormonal regulation in response to environmental stimuli. The MAPK signaling pathway further highlights its role in transducing stress signals into adaptive cellular responses (see Figure 4). Metabolic pathways, such as carbon metabolism, starch, sucrose metabolism, and phenylpropanoid biosynthesis, are significantly enriched, emphasizing the importance of energy production, carbohydrate utilization, and secondary metabolite synthesis for structural and defensive functions. Stress-related pathways, including glutathione metabolism, peroxisomes, and oxidative phosphorylation, reveal mechanisms for maintaining redox balance and energy homeostasis. Additionally, enrichment in photosynthesis and photosynthesis antenna proteins suggests regulation of light harvesting and energy capture (Appendix A).

### 2.5. Differential Regulation of ABA Metabolites in LcTprxII Overexpression

To investigate the role of *LcTprxII* in hormone signaling, we quantified the levels of key hormones involved in the stress response and growth regulation (Figure 5A), including ABA levels in WT and OE lines under controlled and treated conditions. Under control conditions, no significant difference in ABA content was observed between WT and OE lines (*p* > 0.05). However, following treatment, the OE line exhibited significantly lower ABA accumulation than the WT line (*p* < 0.001). Specifically, ABA levels in the WT treated plants remained comparable to control levels, whereas the treated OE line showed a marked reduction. ABA-GE, a conjugated form of ABA, displayed a distinct pattern (Figure 5B). Under control conditions, the ABA-GE content did not differ significantly between the WT and OE lines (*p* > 0.05). Similarly, no significant difference was found between WT and OE lines after treatment (*p* > 0.05). However, the ABA-GE content in both the OE lines and the WT plants under control and treatment conditions was insignificant. Furthermore, the levels of GA19 and GA20, two precursors of the gibberellin biosynthesis pathway, were examined. Under control conditions, GA19 levels were similar between WT and OE lines (*p* > 0.05) (Figure 5C). Following treatment, the OE line exhibited a notable increase in GA19 content compared with the treated WT line (*p* < 0.05). While the treated WT line showed a slight, non-significant decrease in GA19, the OE line showed a trend towards increased GA19. In contrast, the GA20 response presented a significant difference between groups (Figure 5D). Under control conditions, GA20 levels were not significantly different between the WT and OE lines. However, after treatment, the OE line showed a considerable increase compared to the WT group (*p* < 0.01).

### 2.6. Difference in Agronomic Traits Due to LcTprxII Overexpression

To investigate whether the results obtained in the greenhouse were reproducible under field conditions, homozygous transgenic and control plants were evaluated in a field trial. The field was established in GMO fields in Jilin province, China (125° E, 44° N), following three or more biological replications in randomized blocks per line (transgenic, control), respectively. The results were significantly higher in the transgenic maize plants under alkaline conditions with regard to parameters like plant height, ear height, grain weight, and cob diameter, while #1 and #2 exhibited an increase in plant height of 23.93%, 20.07%, and 21.16% and an increase in ear height of 31.27%, 33.10%, and 34.87%, respectively, compared with the wild type (Appendix A). Moreover, interestingly, we found that the plant height improvement was effective in transgenic lines under alkaline stress compared with the WT plant height. Furthermore, the agronomic trait between transgenic lines and WT was investigated. Transgenic plants achieved a significant increase in ear height, grain weight, and cob diameter (Appendix A). The results of this study indicated that the transgenic lines of the LcTprxII gene express resistance to alkaline tolerance in maize.

### 2.7. Differentially Expressed TFs Under Alkaline (Na_2_CO_3_) Stress

To investigate the potential functions of the *LcTprxII-OE* gene, transcriptome analysis revealed significant differential expression of several transcription factor families in response to stress treatment in maize. The major families included 25 DEGs encoding vimyb avian myeloblastosis viral oncogene homolog (MYB), 28 coding for domain-containing protein (WRKY), 13 coding for NAC domain-containing proteins (NAC), 7 coding for bZIP basic leucine zipper (bZIP), 2 coding for ethylene-responsive factor (ERF), 4 coding for heat stress factor (HSF), and 16 coding for basic helix-loop-helix (bHLH). Among these, 95 differentially expressed genes (DEGs) showed consistent expression patterns in transgenic overexpression wild type (Appendix A). However, DEGs of these families showed different expression patterns; there are 5 upregulated and 20 downregulated DEGs for MYB, 4 upregulated and 24 downregulated DEGs for WRKY, 2 upregulated and 11 downregulated DEGs for NAC, 3 upregulated and 4 downregulated DEGs for bZIP, 0 upregulated and 2 downregulated DEGs for AP2/ERF, 1 upregulated and 3 downregulated DEGs for HSF, and 8 upregulated and 9 downregulated DEGs for bHLH.

### 2.8. The Verification of Gene Expression Candidates

To verify the integrity of the RNA-Seq data, three abscisic acids (ABAs) and six DEGs were selected from each TF family, and qRT-PCR also verified the expression patterns. Zm00001eb396390 (MYB), Zm00001eb419370 (WRKY), Zm00001eb121380 (NAC), Zm00001eb212940 (bZIP), Zm00001eb124740 (AP2/ERF), and Zm00001eb239380 (HSF) were upregulated in *LcTprxII-OE* maize, while they were downregulated in *LcTprxII*-OE maize and randomly chosen for qRT-PCR (Appendix A). These results are consistent with those of the transcriptome data (Figure 6B and Table 1). Therefore, this indicates that these TFs play an important role in the tolerance of transgenic maize to drought stress.

## 3. Discussion

Type II peroxiredoxins (Prxs) play a crucial role in regulating plant physiological processes, serving as key modulators of growth and development mechanisms. Plants must navigate complex challenges involving metabolic adaptations, protective responses, and reproductive strategies. Environmental stressors like alkaline conditions can significantly impair plant function by triggering ROS accumulation and undermining cellular stress resilience [24]. To explore the genetic modification of *LcTprxII* overexpression in transgenic maize, it provides a molecular strategy for enhancing alkaline stress tolerance, revealing intricate mechanisms of plant adaptive responses. Interestingly, protein alignment indicated that *LcTprxII* and its homologs contain an N-terminal secretory signal peptide (Appendix A). These results supported the idea that *LcTprxII* is secreted and active in type II peroxiredoxin (Figure 1A). Different types of peroxiredoxin were induced by differential stress and have been characterized by alkaline stress [25].

Phylogenetic analysis showed that *LcTprxII* is a homolog of several reported type II peroxiredoxin (PrxsII) proteins, including OsPrxIIC, ZmPrx5, AtPrxIIB, AtPrxIIC, and AtPrxIID [26,27]. These PrxsII proteins were proposed to be involved in plant defense responses. The physiological and biochemical indicators and transcriptome analysis revealed the remarkable role of LcTprxII overexpression in transgenic maize for the enhancement of alkaline tolerance (Figure 3). This research indicated that overexpression of *LcTprxII* in transgenic maize significantly improves alkaline stress tolerance through various mechanisms, including enhanced antioxidant enzyme activity, improved stress signaling, and transcriptional regulation. Under alkaline stress, the transgenic plants showed increased activity of APX, SOD, CAT, and POD, which collectively reduced ROS levels and limited membrane lipid peroxidation (Figure 3C–F). These results are consistent with the established role of type II peroxiredoxins in strengthening antioxidant defenses and promoting stress resilience [25,26,27]. Importantly, homologous proteins, such as ZmPrx5-1 in maize and OsPrxII2 in rice, have also been shown to enhance ROS scavenging and maintain cellular metabolism under various stress conditions [28], emphasizing the conserved function of PrxII proteins across plant species.The transcriptome analysis revealed that the DEGs were significantly enriched in the plant hormone signal transduction pathways and MAPK signal pathways (Figure 4A,B).

Transcriptomic profiling revealed significant differential expression of genes associated with hormone signaling, particularly in the abscisic acid (ABA) and mitogen-activated protein kinase (MAPK) pathways. ABA signaling, known for regulating ion homeostasis and metabolic adjustments under stress [29,30,31], was enriched in transgenic lines. This included genes encoding PYR/PYL/RCAR receptors, which modulate downstream stress responses. In Figure 7, ABA signaling involves PYR/PYL receptors that inhibit PP2C phosphatases, leading to the activation of SnRK2 kinases. These kinases then phosphorylate downstream targets, such as AREB/ABF transcription factors and SLAC1, which regulate stomatal closure and stress-responsive gene expression. Additionally, MAPK pathway activation, critical for amplifying stress signals [32], was observed, suggesting a synergistic interaction between ABA and MAPK signaling. Alkaline stress likely induces ABA-mediated PYR expression, which relieves SnRK2 inhibition and activates MAPK cascades (ZmMAPKKK18, homologous to drought-responsive NPK1 in tobacco) [25]. This sequential activation enhances stress signal transduction, as evidenced by the rapid upregulation of Zm00001d043742 (50-fold at 6 h), an ortholog of rice OsMAPKKK63 implicated in salt tolerance [33,34]. The MAPK pathway, depicted in Figure 7 as a cascade involving MAPKKK, MAPKK, and MAPK, amplifies stress signals and interacts with ABA signaling to enhance stress responses. This synergistic interaction is critical for the plant’s ability to adapt to alkaline stress. By integrating the roles of ABA and MAPK signaling pathways, our findings highlight the complex regulatory networks that plants utilize to respond to environmental stresses. The interplay between these pathways not only enhances stress signal transduction but also ensures a coordinated response that mitigates the adverse effects of alkaline stress. This integrated approach provides a comprehensive understanding of the molecular mechanisms underlying stress tolerance in transgenic maize overexpressing *LcTprxII*.

Plants have evolved a range of stress resistance mechanisms in which transcription factors play a key regulatory role. Although *LcTprxII* is a peroxiredoxin, it likely regulates TF genes indirectly by altering the redox status of the cell. This modulates ROS levels, which are known to affect transcriptional responses. TFs linked to stress adaptation were differentially expressed in transgenic plants (Table 1).

Upregulated MYB (Zm00001eb396390) and WRKY TFs (Zm00001eb419370, Zm00001eb149570) align with their roles in oxidative stress mitigation and defense [34]. Conversely, the downregulation of AP2/ERF (Zm00001eb124740) implies a potentially negative regulatory role in stress responses [43]. The divergent expression of heat shock factors (HSFs) in upregulated Zm00001eb239380 versus downregulated Zm00001eb198620 further highlights the functional complexity within TF families during stress [44]. These findings mirror observations in drought-tolerant maize [45,46], suggesting conserved TF networks across stress types. Critically, transgenic maize maintained robust growth and yield in natural alkaline soils, in contrast to prior studies that were limited to controlled environments [47]. This field validation underscores the translational potential of *LcTprxII* overexpression for breeding alkaline-resistant crops. The absence of growth penalties under field conditions further supports its agricultural applicability. Although our work elucidates key mechanisms, several functional characterizations of specific differentially expressed genes (DEGs) are needed to dissect their roles in alkaline tolerance. Additionally, exploring ABA, MAPK, and other hormone pathways (e.g., auxin and ethylene) could unravel broader regulatory networks. Extended field trials across diverse alkaline environments will validate the universality of these findings and assess the stability of *LcTprxII*-mediated tolerance under various conditions. In addition, LcTprxII overexpression is expected to modulate ROS levels, thereby impacting redox-sensitive TF activation, hormone (ABA) signaling, and MAPK kinase cascades. These interconnected pathways collectively shape the differential TF expression observed in the transgenic maize. Such a mechanism aligns with reports that peroxiredoxins regulate H_2_O_2_-dependent transcription and ABA responses.

## 4. Materials and Methods

### 4.1. Cloning and Transformation of LcTprxII

In this study, the *LcTprxII* gene, which confers alkaline resistance, was previously cloned from *Leymus chinensis* [48]. The LcTprxII gene (GenBank accession no GQ397275) was inserted into the monocot expression vector pCAM3300, resulting in the construct designated as pCAM3300-LcTprxII. The expression of the LcTprxII gene was driven by the ubiquitin promoter (Figure 1A). Transgenic maize plants were generated through Agrobacterium-mediated transformation, following the methodology described by [49]. The *LcTprxII* gene (T_0_) was introduced into the inbred maize through backcrossing. BC_1_F_1_ was produced, BC_1_F_1_ was backcrossed to produce BC_2_F_1_, and this process was repeated until BC_4_F_1_ lines were generated. Transgenic plants were selected based on herbicide resistance and PCR confirmation of the *LcTprxII* gene. Four independent transgenic lines were used for further analysis.

### 4.2. Phenotyping and Relative Chlorophyll Content Analysis

Plant height, measured from the first internode to the tip of the first leaf, was recorded starting from the 3-leaf stage in seedlings grown in the greenhouse, with three biological replicates. The relative chlorophyll content was assessed using a SPAD-502 meter (Konica Minolta, Chiyoda City, Japan) on the third fully expanded leaf, also with three replicates.

### 4.3. Physiological and Biochemical Analysis

Frozen leaf tissues (1 g) were homogenized in 10 mL of 0.1 M potassium phosphate buffer (pH 7.0) containing 0.1 mM EDTA-Na_2_, 0.5 mM ascorbate, and 1% polyvinyl polypyrrolidone (PVPP). After filtration, the homogenate underwent centrifugation at 28,710× *g* for 10 min while maintained at 4 °C. Protein content and antioxidant enzyme activities were analyzed using the supernatant. Following established protocols, nitroblue tetrazolium (NBT) staining was performed on leaves collected from both wild-type (‘WT’) and overexpression (‘OE’) seedlings that had been exposed to saline–alkaline stress conditions [50]. The H_2_O_2_ concentration was calculated using a standard curve [51]. Superoxide dismutase activity was determined according to [52]. Peroxidase activity was determined according to [53]. Catalase activity was assayed using the method described by [54]. Ascorbate peroxidase activity and MDA content were determined according to [55,56]. The biological data were used for all measurements, and statistical differences were determined using Student’s two-tailed *t*-test.

### 4.4. Determination of Endogenous Plant Hormones (GA, ABA)

The shoot tips from vegetative stage V3 homozygous mutants and wild-type controls grown in the greenhouse were harvested, with ten individual plants combined to create a single biological replicate. The previously mentioned endogenous hormone levels were analyzed using NetWare (http://www.metware.cn, accessed on 17 August 2024) on the AB Sciex QTRAP 6500 LC-MS/MS platform. The procedures for the analysis are outlined as follows: for the determination of endogenous gibberellins (GAs), refer to the outlined method [57]; for abscisic acid (ABA) determination, refer to the specified method [58].

### 4.5. Field Trials and Yield Assessment of LcTprxII Overexpression

Transgenic LcTprxII plants and wild-type controls were grown in genetically modified organism (GMO) fields in Jilin province, China (125° E, 44° N), following a randomized block design with two replications for each genotype. Each block comprised 40 plants arranged in two rows of 20, with 25 cm × 30 cm spacing between each plant. In general, two transgenic plants of #1 and #2 were selected to be planted in the field; the #4 transgenic was not selected due to its poor performance at the seedling stage. The plants were subjected to watering and alkaline stress treatments. Morphological and yield parameters were assessed, with plant height measured from the ground to the tassel tip and ear height measured from the ground to the base of the first ear. Stem diameter was measured at the first internode above ground using a Vernier caliper oriented perpendicular to the direction of leaf growth. The harvest index (HI) was calculated as the seed weight ratio to total biomass, including dry vegetative tissue and panicle weight. Unit yield was obtained by extrapolating seed weight from the block area to a unit area. All measurements were conducted on at least 24 plants per maize line, excluding those from the two outermost rows in each block.

### 4.6. RNA-Seq Analysis and Expression Patterns of LcTprxII-OE

RNA sequencing was conducted on LcTprxII-OE (overexpression) seedlings and wild-type seedlings at the V3 developmental stage. Leaf tissues were harvested, and RNA was extracted from transgenic overexpression and wild-type plants and divided into three biological replicates. Sequencing libraries were generated using the NEBNext UltraTM RNA Library Prep Kit for Illumina (NEB, Ipswich, MA, USA). Clean reads were mapped to the maize reference genome (http//ftp.ensemblgenomes.org/pub/plants/release-32/fasta/zea_mays/, accessed on 17 August 2024) using the Tophat2 algorithm, which was employed to process the data. Sequences exhibiting exact matches or containing a single mismatch with the reference genome underwent additional examination and classification according to two reference databases: the KEGG Orthology database (https://www.kegg.jp/kegg/ko.html, accessed on 17 August 2024) and the Gene Ontology (GO) database (http://www.geneontology.org/, accessed on 17 August 2024). Gene expression was measured as fragments per kilobase of the transcript per million mapped fragments. Statistical significance was assessed using q-values for *p*-value adjustment following the method of Storey and Tibshirani [59]. Differentially expressed genes were identified using dual criteria: a Q-value below 0.005 and a log2 (fold change) equal to or exceeding 1. The raw sequencing data have been uploaded to the NCBI public database under project number PRJNA1222582.

### 4.7. Real-Time PCR (qRT-PCR) Analysis

Total RNA was extracted using the TRIzol reagent from Sangon Biotech, Shanghai Co., Ltd. For cDNA synthesis, 500 ng of DNase-treated RNA was utilized with an RT reagent kit from TakaRa (Shanghai, China), following the manufacturer’s instructions. Quantitative reverse transcription PCR (qRT-PCR) was conducted on an ABI7500 qRT-PCR System using the SYBR^®^ RT-PCR Kit from Takara (Dalian, China). The maize actin1 gene served as the internal control. Specific primer sequences for the LcTprxII gene were designed for the qRT-PCR assays. The thermal cycling conditions included an initial denaturation at 95 °C for 5 min, followed by 40 cycles of 95 °C for 15 s (denaturation), 54–62 °C for 15 s (annealing), and 72 °C for 37 s (extension). A mixed solution without a template was included as a negative control. Statistical analyses were performed using the 2^−∆∆CT^ [58,60] Method. The qRT-PCR experiments were conducted with three biological replicates. Gene expression pattern analysis was performed by using qRT-PCR. To verify the RNA sequencing, leaves of the WT and transgenic plants at the V3 stage were harvested for qRT-PCR. The actin gene (GenBank accession no AB181991) was used as an internal control. Each experiment had three biological replicates.

### 4.8. Statistical Data Analysis

Statistical analysis was conducted on phenotyping, relative chlorophyll content assessment, alkaline treatment, and physiological/biochemical evaluations. The data were statistically processed, and the means of each transgenic line (#1, #2, #4) and the wild-type control (WT) were compared using a *t*-test, with calculations performed using GraphPad version 10.0 software (San Diego, CA, USA) [61]. Results are presented as the mean accompanied by the standard error (SE) of the mean. All experiments included three or more biological replicates.

## 5. Conclusions

The study shows that LcTprxII overexpression in transgenic maize improves alkaline stress tolerance through enhanced antioxidant defense, hormonal regulation, and transcriptional reprogramming. The transgenic lines showed elevated antioxidant enzyme activities and reduced oxidative damage, with lower malondialdehyde levels. Transcriptomic profiling identified 3733 differentially expressed genes (DEGs) enriching ABA and MAPK signaling pathways. Hormonal analysis revealed reduced ABA and increased GA levels under stress, promoting growth. Field trials confirmed the agronomic potential of *LcTprxII-OE* maize by maintaining yield and biomass under alkaline conditions. These findings suggest LcTprxII as a promising candidate for breeding stress-tolerant crops, particularly in alkaline-soil-affected regions. Future research should focus on the functional characterization of specific DEGs, exploring *LcTprxII* and other hormone pathways, and developing *LcTprxII-OE* lines in other crops.

## Figures and Tables

**Figure 1 plants-14-01467-f001:**
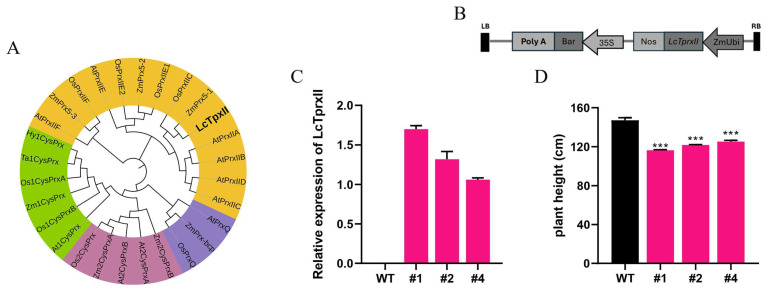
Phylogenetic relationship and expression pattern of LcTprxII. (**A**) Phylogenetic tree relationship with plant Prxs sequences retrieved through BLAST (https://www.ncbi.nlm.nih.gov/, accessed on 5 April 2025) in the NCBI database. The accession numbers, types, and species of Prxs from different plants are given. The bar indicates the scale for branch length. (**B**) The map of the pCAMBIA3300-T vector. (**C**) qRT-PCR was used to measure *LcTprxII* expression levels across different transgenic lines. The analysis included three biological duplications, with each biological duplication comprising three technical replicates. (**D**) Plant heights of WT and different transgenics. Results are expressed as mean values ± standard error (SE). Significant differences between wild-type (WT) plants and transgenic lines (#1, #2, and #4) are indicated by asterisks (*t*-test, *** *p* < 0.001).

**Figure 2 plants-14-01467-f002:**
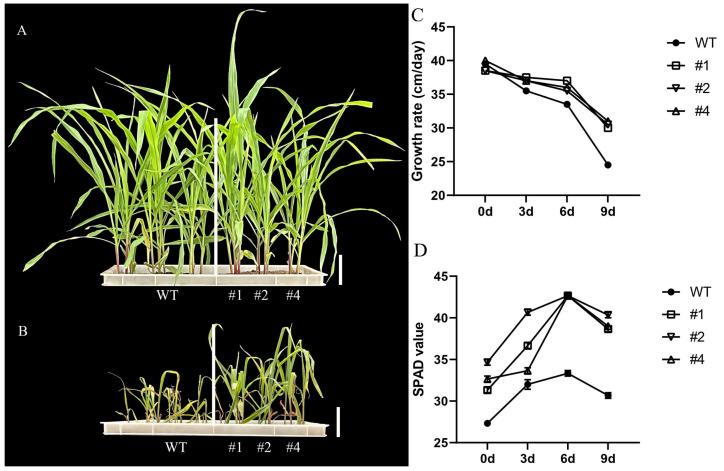
The response of overexpression of *LcTprxII* when subjected to alkaline conditions was evaluated. (**A**) The growth performance of WT and transgenic plants before alkaline stress, (**B**) the growth performances of WT and transgenic plants after 9 days of alkaline stress, (**C**) the growth rate of WT and transgenic plants, and (**D**) chlorophyll SPAD values of the third expanded leaves of WT and transgenic lines. WT and transgenic plants were grown in a greenhouse until the leaf stage and treated with 75 mMol of Na_2_CO_3_. The values were measured during the 9 days of alkaline treatment. Bar = 10 cm. The data are presented as mean ± SE (*n* = 3).

**Figure 3 plants-14-01467-f003:**
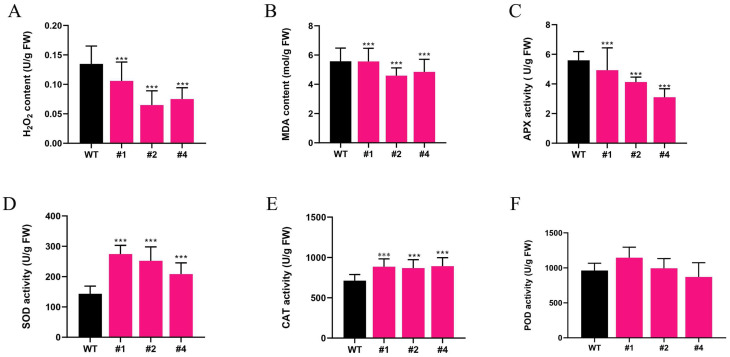
Comparison of H_2_O_2_ content, CAT content, and antioxidant enzyme activity in transgenic maize and WT under alkaline (Na_2_CO_3_) treatment. WT and LcTprxII transgenic plants were grown in a greenhouse for 9 days. H_2_O_2_ content (**A**), CAT content (**B**), MDA (**C**), APX (**D**), SOD (**E**), and POD (**F**) were measured in WT and transgenic plants after 9 days of alkaline stress. Results are presented as mean values ± SE (*n* = 3). Asterisks denote significant differences between the wild-type (WT) and transgenic lines (#1, #2, and #4), as determined through the *t*-test (*** *p* < 0.001).

**Figure 4 plants-14-01467-f004:**
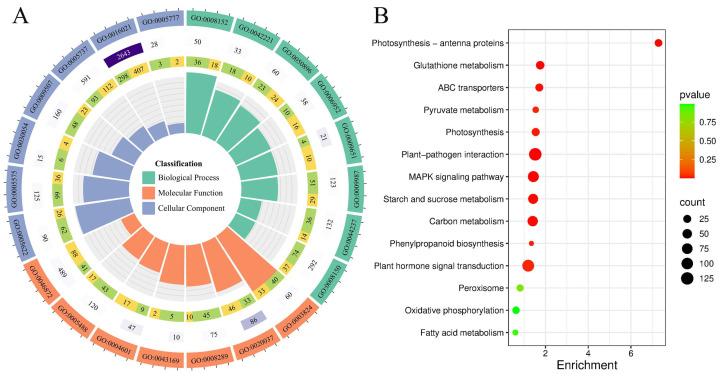
The enrichment of DEGs in the Biological Process GO terms and KEGG pathways analysis. The enrichment of DEGs in the biological GO terms: GO:0005737: cytoplasm; GO:0005622: intracellular; GO:0009507: chloroplast; GO:0005575: cellular_component; GO:0016021: integral component of the membrane; GO:0005777: peroxisome; GO:0030054: cell junction; GO:0044237: cellular metabolic process; GO:0008150: biological_process; GO:0008152: metabolic process; GO:0009987: cellular process; GO:0050896: response to stimulus; GO:0042221: response to chemical; GO:0009651: response to salt stress; GO:0006952: defense response; GO:0043169: cation binding; GO:0008289: lipid binding; GO:0020037: heme binding; GO:0046872: metal ion binding; GO:0005488: binding; GO:0003824: catalytic activity; GO:0004601: peroxidase activity. (**A**) The enrichment of DEGs were divided into three categories of biological process, cellular components, and molecular function with GO analysis. (**B**) KEGG pathway enrichment analysis.

**Figure 5 plants-14-01467-f005:**
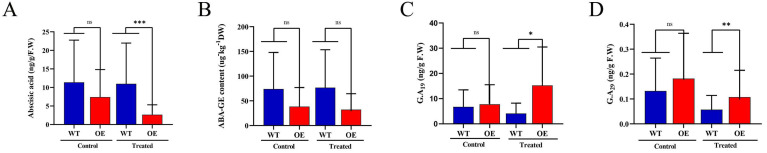
Determinations of plant hormone in a three leaf stage overexpression LcTprxII treated with 75 mmol/L alkaline stress. (**A**) The Abscisic acid (ABA) of WT and overexpression *LcTprxII* treated with water and 75mmol/L alkaline stress for 9 days, (**B**) ABA-GE content of WT and overexpression *LcTprxII* treated with water and 75 mmol/L alkaline stress for 9 days, (**C**) Gibberellin A19 (GA19) WT and overexpression LcTprxII treated with water and alkaline stress for 9 days, (**D**) Gibberellin A29 (GA29) WT and overexpression *LcTprxII* treated with water and alkaline stress for 9 days. The data were presented as mean ± SE. The asterisks indicate the signifcant diferences between overexpression *LcTprxII* and WT plants (*t*-test, * *p* < 0.5, ** *p* < 0.01, *** *p* < 0.001).

**Figure 6 plants-14-01467-f006:**
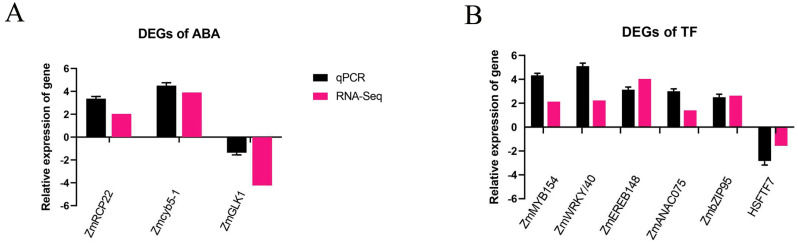
RNA-seq analysis identified the expression patterns of differentially expressed genes (DEGs) associated with abscisic acid (ABA) and transcription factors (TFs) in transgenic maize line #1. (**A**) The relative expression of DEGs in the ABA metabolism pathway. Zm00001eb294600 (*RCP22*), Zm00001eb213190 (*cyb5-1*), Zm00001eb273440 (GLK1). (**B**) The relative expression of DEGs in TFs. Zm00001eb239380 (*HSFTF11*), Zm00001eb121380 (*NACTF94*), Zm00001eb124740 (*EREB148*), Zm00001eb419370 (*MYB36*), Zm00001eb212940 (*bZIP6*), Zm00001eb419370 (*WRKY40*). Each reaction was conducted with three biological duplicates, and every biological duplicate included three technical replicates. Data analysis employed the 2^−ΔΔCT^ method for statistical evaluation. Results are displayed as mean values, with error bars representing the standard error (SE).

**Figure 7 plants-14-01467-f007:**
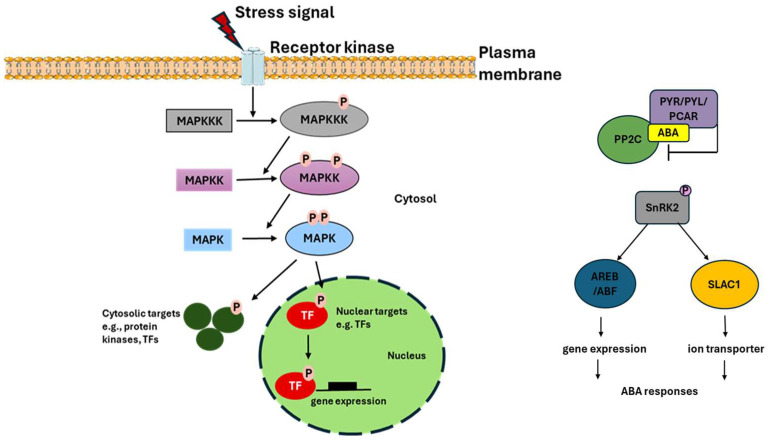
Two key signaling pathways activated in response to stress: the MAPK cascade and the ABA-dependent pathway. The MAPK cascade involves sequential phosphorylation of MAPKKK, MAPKK, and MAPK, leading to transcription factor activation and stress-responsive gene expression, while the ABA pathway involves PYR/PYL/PCAR-mediated inhibition of PP2C, activation of SnRK2, and subsequent regulation of downstream targets, including SLAC1 and transcription factors, to modulate stress adaptation.

**Table 1 plants-14-01467-t001:** DEGs of the transcription factors involved in abiotic stress responses.

Group	Gene ID	Log_2_FC	Swiss_Prot Annotation	NR_Annotation	Reference
MYB	Zm00001eb396390	1.09	Myb-related protein P OS = *Zea mays* OX = 4577 GN = P PE = 2 SV = 1	myb-related protein Myb4 [*Zea mays*]	[35]
WRKY/40	Zm00001eb419370	2.68	WRKY transcription factor SUSIBA2 OS = Hordeum vulgare OX = 4513 GN = WRKY46 PE = 1 SV = 1	WRKY transcription factor SUSIBA2-like [*Zea mays*]	[36]
	Zm00001eb149570	1.24	WRKY DNA-binding transcription factor 70 OS = Solanum lycopersicum OX = 4081 GN = WRKY70 PE = 2 SV = 1	WRKY DNA binding domain-containing protein [*Zea mays*]	[37]
NAC	Zm00001eb121380	1.01	NAC domain-containing protein 75 OS = Arabidopsis thaliana OX = 3702 GN = NAC075 PE = 2 SV = 1	ANAC075 isoform X2 [*Zea mays*]	[38]
bZIP	Zm00001eb212940	1.03	bZIP transcription factor RISBZ2 OS = Oryza sativa subsp. japonica OX = 39947 GN = RISBZ2 PE = 1 SV = 1	unknown [*Zea mays*]	[39]
AP2/ERF	Zm00001eb124740	−1.42	AP2/ERF and B3 domain-containing protein Os01g0141000 OS = Oryza sativa subsp. japonica OX = 39947 GN = Os01g0141000 PE = 2 SV = 1	AP2/ERF and B3 domain-containing protein Os01g0141000 [*Zea mays*]	[36,40]
HSF	Zm00001eb198620	−1.38	Heat stress transcription factor B-2b OS = Oryza sativa subsp. japonica OX = 39947 GN = HSFB2B PE = 2 SV = 1	uncharacterized protein LOC100283530 isoform X1 [*Zea mays*]	[37]
	Zm00001eb239380	1.47	Heat stress transcription factor C-2a OS = Oryza sativa subsp. japonica OX = 39947 GN = HSFC2A PE = 2 SV = 1	Unknown [*Zea mays*]	[41,42]

The first column represents the group name of DEGs involved in abiotic stress responses; the second column represents the gene ID; and the subsequent columns represent the fold change (log2FC) of the DEGs. The last column represents the annotation information of DEGs in Swiss-Prot and NR databases.

## Data Availability

The original contributions presented in this study are included in the article/Appendix A. Further inquiries can be directed to the corresponding author(s).

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
