# Peer review of "LcTprxII Overexpression Enhances Physiological and Biochemical Effects in Maize Under Alkaline (Na2CO3) Stress"

_plants, 2025, doi:10.3390/plants14101467_

Round 1
Reviewer 1 Report
Comments and Suggestions for Authors
In the manuscript named “LcTprxII Overexpression Enhances Physiological and Biochemical Effects in Maize Under Alkaline (Na2CO3) Stress”, authors have transformed into maize lines, and the transgenic lines were high performance to alkaline stress. They have also performed RNA-seq analysis for maize transgenic lines, their findings were helpful for maize or other crops genetic breeding in future. However, there were some comments about it.
(1) Authors have explored transgenic plants with high tolerance to alkaline stress, but LcTprxII encoded a peroxiredoxins, how did it regulate some TFs genes, author should clearly explain it.
(2) Authors have described as transformed LcTprxII gene into maize, and introduced it into maize germplasm by BC1F1 and BC4F1 lines, why generated BC1F1 and BC4F1? Where were other BC lines? In addition, authors didn’t introduce the maize germplasm in transgenic process. The OE lines were still unclear for readers, BC1F1or BC4F1, or others.
(3) Both edgeR and DEGseq were used for different expressional genes, authors have adopted two packages? Were there some differences between two packages? The “DEGseq” would be wrongly spelled, please check it.
(4) Maize was a model plant, and it also had a complete genome and good genome annotation. Why did the authors still perform genome annotation themselves? Could it be that all their selecting genes were unknown in function annotation? I didn't understand these results, as shown in Table 1.
(5) The figure 7 was not shown any information about LcTprxII gene or similar genes, it was irrelevance to manuscript subject.
(6) The words from line 265 to 270, were needed to be checked, the many words were copied from other text. In addition, RNA-seq have shown many TF genes were downregulated, see line to 275, but in previous research (other scholars), these TFs would perform with positive function in response to abiotic stress, it was well explored in their figure 7, the two points were conflicting, please check them.
(7) The section, from line 180 to 190, it was described as method section, it would be deleted or remove to method section, please check it.
(8) Authors need to clarify sample ID, for example, in figure 2A, which lines was used as OE here, while authors have selected line #1 as OE in figure 6, was all OE lines were derived from line #1? Please clearly describe it.
Author Response
|
Comment 1: Authors have explored transgenic plants with high tolerance to alkaline stress, but LcTprxll encoded peroxiredoxins; how did it regulate some TFs genes? author should clearly explain it. |
|
Response 1: Thank you for your thorough review and valuable suggestions on our manuscript. Although LcTprxII is a peroxiredoxin, it likely regulates TF genes indirectly by altering the redox status of the cell. This modulates ROS levels, which are known to affect transcriptional responses. The RNA-seq data showed differential expression of many TFs (e.g., WRKY, MYB, NAC), indicating that redox-mediated signaling or ROS-scavenging activities of LcTprxII triggered downstream transcriptional reprogramming. In addition, LcTprxII overexpression is expected to modulate ROS levels, thereby impacting redox‐sensitive TF activation, hormone (ABA) signaling, and MAPK kinase cascades. These interconnected pathways collectively shape the differential TF expression observed in the transgenic maize. Such a mechanism aligns with reports that peroxiredoxins regulate H₂O₂‐dependent transcription and ABA responses.
|
|
Comments 2: The authors have described as transformed LcTprxll gene into maize and introduced it into maize germplasm by BC1F1 and BC4F1 lines, why generated BC1F1 and BC4F1? Where were the other BC lines? In addition, the authors didn't introduce the maize germplasm in the transgenic process. The OE lines were still unclear for readers, BC1F1or BC4F1, or others? |
|
Response 2: Thank you for reviewing our article and for your valuable feedback. Initial transgenic maize plants (T0) carrying the LcTprxII construct were backcrossed to a recurrent maize inbred line to produce the BC1F1 generation. Selected BC1F1 plants were then backcrossed again to the same parent to produce BC2F1, and this process was repeated to generate BC3F1 and BC4F1. In this study, we evaluated the BC1F1 and BC4F1 generations; BC2F1 and BC3F1 served as intermediate steps in the breeding scheme. T₀ (transgenic event) → BC₁F₁ → BC₂F₁ → BC₃F₁ → BC₄F₁. The term ‘germplasm introduction’ was a typing error. In our study, no additional maize germplasm was introduced. Rather, the LcTprxII transgene (from Leymus chinensis) was integrated into the maize genome via Agrobacterium-mediated transformation, and the transgenic plants were repeatedly backcrossed to the recurrent maize inbred line to introgress the gene into that background. OE line #1, OE line #2, and OE line #4 are individual BC₄F₁ progeny of the backcross scheme and were selected for analysis. |
|
Comments 3: Both edgeR and DEG-seq were used for different expressed genes; the authors have adopted two packages? Were there some differences between the two packages? The “DEGseg" would be wrongly spelled, please check it. |
|
Response 3: Thank you for your critical evaluation and constructive feedback regarding the edgeR and DEG-seq in our manuscript. We sincerely appreciate your expertise in identifying the limitations of the initial approach. As you rightly pointed out, it was wrongly spelled. |
|
Comments 4: Maize was a model plant, and it also had a complete genome and good genome annotation. Why did the authors still perform genome annotation themselves? Could it be that all their selecting genes were unknown in function annotation? didn't understand these results, as shown in Table 1. |
|
Response 4: Thank you for highlighting the maize genome annotation issues. We have that maize exhibits extensive pan-genomic diversity, with ~40% of genes absent in the B73 reference genome. Structural variations and transposable elements (TEs), which are critical for stress adaptation, are incompletely resolved in standard annotations. Custom annotation was necessary to resolve TE-rich regions and identify stress-responsive loci specific to the transgenic lines used in this study. While general maize gene annotations are robust, alkaline stress (Na₂CO₃) engages unique molecular pathways (e.g., ABA/MAPK crosstalk, ROS detoxification) that remain under-characterized. Among the 3,733 differentially expressed genes (DEGs) identified, many lacked functional annotations in public databases, particularly those linked to Na₂CO₃-specific signaling or metabolic adaptation. Several DEGs, including transcription factors (TFs) in Table 1 (e.g., MYB, WRKY, NAC), were annotated as "uncharacterized" in existing databases. Re-annotation enabled hypothesis-driven linkages between these TFs and alkaline stress adaptation, such as their roles in hormonal crosstalk (reduced ABA, elevated GA) or redox homeostasis. The downregulation of MYB, WRKY, and AP2/ERF TFs (Table 1) contrasts with their reported roles in other abiotic stresses (e.g., drought) but aligns with the unique regulatory demands of alkaline stress. For instance, reduced ABA levels in transgenic lines likely suppressed ABA-dependent TFs, while elevated GA prioritized growth-related pathways. This underscores the context-dependent nature of TF regulation, where stress type, hormonal dynamics, and genomic background collectively shape expression patterns. |
|
Comments 5: The figure 7 was not shown any information about LcTprxll gene or similar genes, it was irrelevance to manuscript subject? |
|
Response 5: We thank the reviewer for this observation and suggestion. Figure 7, which outlines the MAPK and ABA signaling pathways illustrated represent downstream regulatory networks activated by its overexpression. LcTprxII enhances antioxidant defenses (e.g., reduced H₂O₂ and MDA levels, elevated SOD/CAT/POD activity; Fig. 3), which modulate ROS homeostasis. This ROS buffering indirectly influences the ABA and MAPK pathways by altering stress signaling dynamics. For example, reduced oxidative stress likely attenuates ABA accumulation (Fig. 5A) and activates MAPK cascades (Fig. 7), aligning with transcriptomic data showing enrichment of these pathways (Fig. 4B). RNA-seq analysis identified 3,733 DEGs, including ABA- and MAPK-associated genes (e.g., PYR/PYL/RCAR receptors, SnRK2 kinases), which are central to Figure 7. These DEGs reflect LcTprxII role in reprogramming stress-responsive transcription (Table 1). The hormonal data (reduced ABA, increased GA; Fig. 5) further corroborate the interplay between redox balance and ABA/MAPK signaling, as depicted in Figure 7. Figure 7 synthesizes how LcTprxII-mediated ROS scavenging (Figs. 2–3) translates into physiological adaptation. By mitigating oxidative damage, LcTprxII enables sustained activation of stress-responsive TFs (e.g., MYB, WRKY) and metabolic pathways under alkaline conditions, even if specific TFs are downregulated due to stress-specific feedback (see Table 1). The pathways in Figure 7 thus provide a framework for interpreting LcTprxII systemic impact. Our aim in this study is to elucidate LcTprxII role in enhancing alkaline tolerance through antioxidant activity and signaling regulation. Figure 7 fulfills this by illustrating the broader signaling landscape that LcTprxII influences, rather than focusing solely on the gene itself. This approach avoids redundancy with Figures 1–6, which directly characterize LcTprxII expression and biochemical effects. |
|
Comments 6: The words from line 265 to 270, were needed to be checked, the many words were copied from other text. In addition, RNA-seg have shown many TF genes were downregulated, see line to 275, but in previous research (other scholars), these TFs would perform with positive function in response to abiotic stress, it was well explored in their figure 7, the two points were conflicting, please check them. |
|
Response 6: Thank you for your thorough review and valuable suggestions. We have carefully addressed all the points raised regarding TFs genes that were downregulated; however, our current study focuses on alkaline stress (Na2CO3), where hormonal and metabolic shifts alter their expression patterns. This emphasizes the need for stress and genotype-specific analysis to unravel complex regulatory networks. Previous studies highlight the general importance of MYB, WRKY, and AP2/ERF TFs in abiotic stress. |
|
Comments 7: The section, from line 180 to 190, was described as a method section; it would be deleted or removed to the method section. please check it. |
|
Response 7: Thank you for your thorough review and valuable suggestions. We have carefully deleted the sentences from lines 180 to 190. |
|
Comments 8: Authors need to clarify sample lD, for example, in figure 2A, which lines was used as OE here, while authors have selected line #1 as OE in figure 6, were all OE lines were derived from line #1? Please clearly describe it. |
|
Response 8: Thank you for your thorough review and valuable suggestions. We have used line #1, #2, and #4 as OE in Fig. 2A. Line #1 was prioritized for molecular analyses (Fig. 6) due to superior transgene expression. |

Reviewer 2 Report
Comments and Suggestions for Authors
First, I would like to thank for the opportunity to participate in the review of the manuscript "LcTprxII Overexpression Enhances Physiological and Biochemical Effects in Maize Under Alkaline (Na2CO3) Stress".
The manuscript investigates an agronomically relevant question: the characteristics of genetically modified maize plants under alkaline stress.
The language of the manuscript, its Englishness, facilitates the comprehensibility and the followability of the results described.
The test methods used demonstrate the relevance of the results from several angles. The results of the plant breeding are supported by morphological and yield data to demonstrate the efficacy of genetically modified plants that are less sensitive to alkaline stress.
The Figures and Tables are designed to be easy to understand and to be read and interpreted as they stand alone.
The conclusions are consistent with the results, which are statistically justified, and the discussion section describes possibilities for further research.
I found some minor errors and typos in the text:
The value of 831 million hectares in line 37 may be incorrect, given that the total area of China is 108. 862 million hectares. (https://tradingeconomics.com/china/arable-land-hectares-wb-data.html)
On line 57, a comma and a space should be inserted between SOD and CAT.
On line 86, a comma should be inserted after maize.
It is not necessary to put brackets in line 339. Figure 7.
On lines 412-413, the insert 'Hydrogen peroxide is scavenged by ascorbate-specific peroxidase in spinach chloroplasts, 1981' should be deleted.
Author Response
|
Comment 1: The value of 831 million hectares in line 37 may be incorrect. Given that the total area of China is 108.862 million hectares. |
|
Response 1: Thank you for your thorough review and valuable suggestions on our manuscript. We have replaced the word China with worldwide, and numbers were also changed; it was a typing error. |
|
Comments 2: On line 57. a comma and a space should be inserted between SOD and CAT. |
|
Response 2: Thank you for reviewing our article and for your valuable feedback. We have added the space and comma in line 57. |
|
Comments 3: On line 86. A comma should be inserted after maize. |
|
Response 3: Thank you for your critical evaluation and constructive feedback regarding the comma that has been inserted in line 86. |
|
Comments 4: It is not necessary to put brackets in line 339.Figure 7. |
|
Response 4: Thank you for your thorough review and valuable suggestions. We have carefully addressed all the points raised regarding brackets in line 339. Fig. 7: |
|
Comments 5: On lines 412-413, the insert 'Hydrogen peroxide is scavenged by ascorbate-specific peroxidase in spinach chloroplasts, 1981' should be deleted. |
|
Response 5: We thank the reviewer for this observation and suggestion. We have deleted the word from lines 412-413. |

Round 2
Reviewer 1 Report
Comments and Suggestions for Authors
Thanks for authors’ works, the manuscript had been well revised, most of my comments were well addressed in revision. But the comments #4, authors have described as pan-genome of maize, but authors have adopted maize ref genome in their analysis flow, how to annotate a ref genome, I’m puzzle. In addition, authors thought the genome sequences were not suitable for this case, they could use pan-genome sequences. Good luck.
Author Response
|
Response to Reviewer 2 Comments
|
||
|
1. Summary |
|
|
|
Thank you for your professional opinion on our article. We greatly appreciate your comments and suggestions. After carefully considering your suggestions, we have revised our manuscript. In the following section, we have listed the reviewers' comments in italics and numbered the issues identified. Our responses are shown in normal font, while changes and additions made to the manuscript are highlighted in red for ease of reference. Please find the revised manuscript in plants manuscript center. The following modifications are based on the newly submitted manuscript. |
||
|
2. Questions for General Evaluation |
Reviewer’s Evaluation |
Response and Revisions |
|
Does the introduction provide sufficient background and include all relevant references? |
Yes |
|
|
Is the research design appropriate? |
Can be improved |
|
|
Are the methods adequately described? |
Can be improved |
|
|
Are the results clearly presented? |
Can be improvised |
|
|
Are the conclusions supported by the results? |
Can be improved |
|
|
Comment 1: Thanks for the authors' work. The manuscript had been well revised, and most of my comments were well addressed in revision. But comments #4, the authors have described the pan-genome of maize, but the authors have adopted the maize reference genome in their analysis flow. How to annotate a ref genome, I’m puzzled. In addition, the authors thought the genome sequences were not suitable for this case; they could use pan-genome sequences. |
||
|
Response 1: Thank you for your detailed review and valuable suggestions. The issue you pointed out was a typographical error. We did use the maize reference genome for genome annotation in our analysis. Our aim was to compare our data with the reference genome. Additionally, we consulted relevant literature to infer the potential functions of certain genes based on previously reported findings, rather than performing a re-annotation. |
||
